# Preparation and Characterization of Semi-Flexible Substrates from Natural Fiber/Nickel Oxide/Polycaprolactone Composite for Microstrip Patch Antenna Circuitries for Microwave Applications

**DOI:** 10.3390/polym12102400

**Published:** 2020-10-19

**Authors:** Ahmad Fahad Ahmad, Sidek Ab Aziz, Yazid Yaakob, Ammar Abd Ali, Nour Attallah Issa

**Affiliations:** 1Department of Physics, Faculty of Science, Universiti Putra Malaysia, Serdang 43400, Malaysia; yazidakob@upm.edu.my; 2Ministry of Education, Babylon 51001, Iraq; ammar111122@gmail.com; 3Department of Physics, Faculty of Science, Universiti Kebangsaan Malaysia, Bangi 43600, Selangor, Malaysia; noortamimi1987@yahoo.com

**Keywords:** nickel oxide, patch antenna, polycaprolactone, CSTMWS, S-parameters

## Abstract

The study intended to utilizing waste organic fiber for low-cost semi-flexible substrate fabrication to develop microstrip patch antennas for low band communication applications. All the semi-flexible substrates (12.2 wt. % OPEFF/87.8 wt. % PCL, 12.2 wt. % NiO/87.8 wt. % PCL, and 25 wt. % OPEFF/25 wt. % NiO/50 wt. % PCL) were fabricated by oil palm empty fruit fiber (OPEFF) mixed with nickel oxide (NiO) nanoparticles reinforced with polycaprolactone (PCL) as a matrix using a Thermo Haake blending machine. The morphology and crystalized structure of the substrates were tested using Fourier transform infrared (FTIR) spectrometry, X-ray diffraction (X-RD) technique, and scanning electron microscopy (SEM), respectively. The thermal stability behavior of the substrates was analyzed using thermogravimetric analysis (TGA) and differential thermogravimetric (DTG) thermogram. The dielectric properties were characterized by an open-ended coaxial probe (OEC) connected with Agilent N5230A PNA-L Network Analyzer included the 85070E2 dielectric software at frequency range of 8 to 12 GHz. The experimental results showed that NiO/OPEFF/PCL composites exhibit controllable permittivity dielectric constant $\varepsilon_{r}^{\prime }\left(f\right)$ between 1.89 and 4.2 (Farad/meter, (F/m)), with loss factor $\varepsilon_{r}^{\prime \prime }\left(f\right)$ between 0.08 and 0.62 F/m, and loss tangent (tan δ) between 0.05 and 0.18. Return losses measurement of the three patch antennas OPEFF/PCL, NiO/PCL, and OPEFF/NiO/PCL are −11.93, −14.2 and −16.3 dB respectively. Finally, the commercial software package, Computer Simulation Technology Microwave Studio (CSTMWS), was used to investigate the antenna performance by simulate S-parameters based on the measured dielectric parameters. A negligible difference is found between the measured and simulated results. Finally, the results obtained encourage the possibility of using natural fibers and nickel oxide in preparation of the substrates utilize at microwave applications.

## 1. Introduction

In recent years, the microstrip patch antenna has been utilized widely in wireless communication technology due to its optimum size and miniature shape characteristics. At present, flexible antennas are used due to their compactness with an advanced flexible substrate that can be readily integrated into electronic devices and cover their functional features. The low-cost flexible antennas are environmentally friendly and easy to fabricate. Several materials with conductive or non-conductive dielectrics properties are used to manufacture this type of flexible antenna [1]. For antenna manufacture, artificial materials are very popular. Therefore, this is prompting for more efforts in intensive research aimed at developing substrates or artificial materials for applications in the areas of electrical and wireless communication such that they have lightweight, ease of fabrication, low cost, less power consumption, and excellent compatibility with planar integrated circuits and even non-planar surfaces [2,3]. Moreover, the organic substrates based on biopolymers composites have been received great attention in both the industry and in academia due to increased concern toward the environmental impact of plastic waste and the saving of limited energy [4,5]. To the development of the substrates that are used at electrical and wireless communication and for the achievement of the cost balance and performance, one or more kinds of materials are incorporated into a single matrix [6]. The hybrid mixtures are often used to improve the structures, dielectric, and mechanical properties of hybrid composites. Where the particles and fibers are evenly and stochastically distributed in hybrid particle/fiber/polymer composites. Since the modulus of inorganic particles is normally much higher than that of polymeric resins, the presence of inorganic particles in the composites would significantly influence the stress transfer from matrix to neighboring fibers [7,8]. 

Natural fibers as a replacement of synthetic fiber have been one of the most popular research topics over the past years due to their environmental and economic benefits. As well, it is due to their inherent properties such as specific strength, renewability, low density, eco-friendly, low cost, sustainable, and abundant availability when compared to synthetic fibers, and their renewable resources absorb carbon dioxide, which alleviates environmental pollution [9,10]. The natural fiber-containing composites are used in the industry field of (automobiles, railway coaches, aero-space), military applications, building and construction industries (ceiling paneling, partition boards), packaging, and consumer products [11]. In recent years, there are several studies shows the possibility of using the natural fibers (carboxy cellulose nanofibers extracted from untreated jute), and fiber with low crystallinity as excellent substrates effective for the removal of heavy metal ions for drinking water purification from poisons like Pb (II) and arsenite (As (III)) component. The results of these studies indicated that this approach exhibited a greatly reduced the need for multi-chemicals, and offered significant benefits in lowering the consumption of water and electric energy when compared with conventional multiple-step processes at bench scale [12,13,14,15]. 

Oil palm empty fruit fiber (OPEFF) is one of the major solid wastes produced by the oil palm industry and has received a lot of interest due to its environmental and economic benefits as a possible replacement for synthetic fibers as reinforcement of various resins for advanced applications [16]. However, the natural fibers suffer from poor mechanical and interfacial properties due to random fiber orientation and weak fiber-matrix interface. Therefore, Sarker et al. [17], have studied how to significantly improves natural fibers mechanical properties by nanoengineered graphene-based natural jute fiber preforms with a new fiber architecture (NFA), by arranging fibers in a parallel direction through individualization and nano surface engineering with graphene derivatives. 

In the microwave circuitry applications, the demand for bio-based plastics has been intensively and continuously increasing due to the fact that the materials are utilized for durable solutions. However, for long term applications that include electronic and microwave circuitry devices, bioplastics have not been quite popular. Through the use of bio-polymer based composites in durable plastic form, the mentioned shortcomings have been greatly mitigated [18,19,20]. Polycaprolactone (PCL) was chosen in this study due to having a high level of flexibility, attractive mechanical properties, lightweight, good dielectric properties, and ability to be easily fabricated and blended [21,22]. In our previous work [23], we prepared and characterized bio-composite consists of OPEFF as a filler hosted in PCL polymer as a matrix. Both real and imaginary relative permittivity values of the samples were measured, it is found that the dielectric properties of the composites were affected by the change in the OPEFF filler percentage at composites in the matrix of the composite. Furthermore, the dielectric constants and loss factors of the composites decreased at an increased frequency. Also, the dielectric properties increased when the percentage of OPEFF filler at the composites increased. Nickel oxide (NiO) semiconductor (p-type) were employed in this study due to their low cost, chemical stability, and excellent properties in magnetic, electrical, and optical, which has a great influence on interaction with electronic fields. Moreover, given its excellent electrochemical, gas sensing, and catalytic properties, it was used widely for electrochromic and solid-state sensors’ applications [24,25]. 

There has been various research reported on electrical applications of nickel oxide-reinforced polymer composites and their electrical properties. The electromagnetic and nanostructure properties of polymer-based materials incorporation with nickel oxide (NiO) in different concentrations of polyvinyl alcohol (PVA) polymer were studied, and the dielectric permittivity values of NiO/PVA composite is found to be higher than for other exhibited samples so it can be introduced as a perfect gate dielectric material [26]. As well as, by adding nickel oxide nanoparticles in different percentages, the dielectric properties of Ba0.7Sr0.3Ti1–xNixO3 (BSTN) ceramics synthesized by solid-state sintering have been studied. It was observed that the dielectric properties (dielectric loss factor and loss tangent) of the ceramics were modified significantly with nickel oxide addition. found these ceramics promising for many applications such as inductor devices, thermistors, batteries, micro-supercapacitors, electrochromic and chemical or temperature sensing devices [27].

The aims of this paper include: Finding the best manufacturing process to prepare the semi-flexible substrates which the copper sheet affixed on both sides to prepare a microstrip patch antennas, the investigation of the electromagnetic theory reliability for the purpose of explaining the behavior of the microstrip patch antennas by assessment of the performance of the antennas, and lastly comparing the simulated results with the measured values.

## 2. Materials and Fabrication Process 

### 2.1. Materials

The materials utilized in this work were oil palm empty fruit fiber (OPEFF) which was supplied by Ulu Langat Oil Palm Mill, Dengkil, Selangor, Malaysia, Polycaprolactone (PCL) (C_6_H_10_O_2_) polymer with 97.0% purity and density of 1.146 g/cm^3^ purchased from Nature Works LLC (Sigma Aldrich, Sarasota, FL, USA) and Nickel Oxide, NiO (99.7% purity) (Sigma Aldrich, Sarasota, FL, USA).

### 2.2. Substrates Fabrication

To prepare the (OPEFF/PCL), (NiO/PCL) and (NiO/OPEFF/PCL) substrates firstly, OPEFF fibers were washed and rinse with warm water at 80 °C for several times to remove all dust and surface wax. Acetone added to the washed fiber to remove any clustered wax and then dried for two hours using a convection oven. By utilizing a Grinding Machine to crush the fiber chains into micro powder size (250 μm) separated by using a mechanical vibration filter [28]. The mixture ratios of the prepared samples (OPEFF/PCL, NiO/PCL, and NiO/OPEFF/PCL) were classified into three cases as listed in Table 1. The prepared samples were mixed carefully using a Thermo Haake blending machine (Thermo Electron GmbH, Karlsruhe, Germany) poly drive three-phase motor set at 1.5 kW, 3.230 V, 40 A and 0–120 rpm drive after heating up the mixture at 80 °C with 50 rpm for 20 min. 

The composite was then preheated for 10 min with an upper and lower plate temperature of 80 °C as a breathing time of two minutes was allowed for the release of bubbles and to reduce void; the composites were then pressed at a pressure of 110 kPa at the same temperature for another 10 min. Finally, the prepared substrate was left to cool to room temperature (27 ℃). The substrate ready for testing as well as for the preparation of (microstrip antenna, and T-junction) for microwave applications. The substrate preparation steps were presented in detail in Figure 1.

### 2.3. Substrates Characterization 

The X-ray diffraction (XRD) technique was adopted to examine the phase composition and crystallinity structure for the prepared specimens. The data were collected using a completely automated Philips X-pert system (PHILIPS PW3040) with a Cu-Kα radiation source of wavelength (λ = 1.5405 Å). The thermal stability of material under test (MUT) that are used in this work (OPEFF, NiO, and PCL), and all the polymer composites were studied by thermogravimetric analysis (TGA) and differential thermogravimetric analysis (DTG) that was conducted using a Perkin Elmer Thermal Analyzer (Perkin Elmer, Waltham, MA, USA) [29]. This analysis was performed at the rate of 10 °C/min with a portion (9–10 mg) of the sample being heated at temperature range of 50–800 °C with nitrogen flowing at the rate of 50 mL/min. After heating, the percentage of residue and weight indicated that weight loss had occurred in the samples. Scanning electron microscopy (SEM) was used to determine the structure of the studied composites with different fillers content. SEM images were captured using a Scanning Electron Microscope SU 6600 (Hitachi High-Tech Corporation, Hitachi, Japan) with field emission gun and accelerating voltage of 20 kV. The distance between the sample and the detector was 6 mm. Samples were placed on double-sided carbon tape on an aluminum holder. All samples were coated by gold with a few nanometers in thickness on sputter coater Quantum Q150T, LOT-Quantum Design (Darmstadt, Germany) prior to the SEM measurements. Each sample was cut with an appropriate size for the study of surface structure. FT-IR spectroscopy is a technique that is sensitive to specific intermolecular interaction where the interaction between the material particles of composites can be measured. Also, the identification of chemical structure and determining the functional groups of compounds while monitoring the shifts in absorption peaks within certain areas can be obtained. The known functional group interaction of the PCL, OPEFF, NiO, and all the composites under test are determined through this monitoring which is done using FT-IR spectroscopy (Perkin Elmer100, Waltham, MA, USA). Prior to scanning, compression of samples into thin film was carried out. Testing was done at room temperature with wavelength ranges of 400 to 4000 cm^−1^. 

### 2.4. Electromagnetic Properties of Substrates 

Specific characterization of OPEFF/PCL, NiO/PCL and NiO/OPEFF/PCL composites was carried out with dielectric and electric properties analysis to study the compatibility of the mixture as well as the change in behavior of the composite with filler, and also dielectric measurement for different composites. The composite will also be used for specific applications. The dielectric characteristics of a composite material depend upon the polarize-ability of its constituents (i.e., matrix and fillers) and are participate mainly by electronic polarizations, atomic, dipoles, and interfacial [30,31]. Basically, the relative permittivity $\varepsilon_{r}\left(f\right)$ is made up of a complex number having real and imaginary parts with frequency dependence as presented in [32].
(1)$$\varepsilon_{r}\left(f\right)=\varepsilon_{r}^{\prime }\left(f\right)-{j\varepsilon }_{r}^{\prime \prime }\left(f\right),$$
where $\varepsilon_{r}^{\prime }\left(f\right)$ represents the dielectric constant of materials, it is the ability to become polarized and to store charge when undergoing an external electric field [33], while $\varepsilon_{r}^{\prime \prime }$ (*f*) is the loss factor and is a measure of how dissipated energy or loss of power of the material when subjected to an external electric field. Another derived parameter used in simulating the antennas is the loss tangent, tan δ is the measure of the electrical energy which can be converted into heat in an insulator [34], which is defined by the equation as follows:(2)$$\tan\delta =\frac{\varepsilon_{r}^{\prime \prime }\left(f\right)}{\varepsilon_{r}^{\prime }\left(f\right)}.$$

The measurements of $\varepsilon_{r}^{\prime }\left(f\right)\;and\;\varepsilon_{r}^{\prime \prime }\left(f\right)$ were carried out with the use of an open ended coaxial dielectric probe kit connected by a coaxial cable with Agilent 8720B, Network Analyzer in the X-band frequency range of 8–12 GHz. 

### 2.5. S-Parameters Measurement

A measurement technique called vector network analyzer-based rectangular waveguide was adopted in the measurement of the two-port network formed S-parameters by placing the studied sample between two waveguides [35]. The overall accuracy of the material properties is dependent on the measurement accuracy of the measured S-parameters. Calibration method called “through-reflect line (TRL)” was employed to eliminate in the measurement, occurring systematic errors [36]. Since a lot of components (such as connectors and cables) are involved in the experimental setup, proper care was taken for assurance of stability of the entire system over the period of measurements. All the measurements reported in this work were carried out at room temperature 

## 3. Microstrip Patch Antennas and Transmission Line Structure Manufacturing

### 3.1. Microstrip Patch Antenna Structure Manufacturing 

In this section, the geometrical details of the patch antenna and manufacturing details are presented. The microstrip antennas are constructed from square patches and square ground planes based on a conducting material such as a copper or gold sheet mounted on the fabricated substrates [37]. The patch antennas are considered with the same dimensions as well as the substrate’s ground planes dimensions and the ground plane is 62 × 62 mm^2^. The copper sheet is made of a copper tape that is stacked on both sides of the fabricated substrates to create a sandwich sheet of copper. The resulting sandwich is pressed less than 3 tons of pressure by a heated platen inside a hot press. The fabricated antennas are fed with 50 Ω SMA ports at the best matching locations from the center of the antenna as presented in Table 2. 

The fabricated antennas NiO/OPEFF/PCL, OPEFF/PCL, and NiO/PCL are presented in Figure 2a–c. The coaxial connector SMA inner conductor extends via the dielectric and connected to the radiating patch by soldering. While to the ground plane, connection is made with the outer conductor. The main advantage of this type of feeding system is that the alimentation can be placed at any desired position inside the patch in order to match with its input impedance. This feed method is easy to fabricate and has low spurious radiation. The other antenna’s structural dimensions are listed in Table 2.

### 3.2. Transmission Line Structure Manufacturing 

The transmission line structure based on T-junction geometry is developed in this section to validate the retrieved complex permittivity obtained from OEC using CSTMWS simulations. As it turned out, the characteristics of the transmission line showed that good agreement had been achieved between the measured and simulated results. To fabricate flexible T-junction circuits and study the RF performances, the Three T-junctions were fabricated using the copper layers (L = 96.33 mm and w = 6 mm and length of T-junction, L = 44.48 mm) based three various substrates such OPEFF/NiO/PCL, OPEFF/PCL, and NiO/PCL as shown in Figure 3. The substrate preparation amount with a thickness of 1 mm for all substrates is illustrated in Table 1 above. The substrate dielectric was 2.8, 3.0 and 3.7 F/m, respectively. Subsequently, the FIT formulations of CSTMWS are conducted to evaluate the S-parameters according to the retrieved constitutive parameters for validations.

## 4. Results and Discussion

### 4.1. X-ray Diffraction

XRD experiments of pure materials and OPEFB/PCL, NiO/PCL, and NiO/OPEFB/PCL composites were performed to determine the influence of NiO and OPEFF filler on the crystalline behavior of the NiO/OPEFB/PCL composites. XRD patterns of all materials under test were scanned in the range of 0°–80° with a 0.02° step size as shown in Figure 4. The different characteristic peak positions of OPEFB powder appear at 2θ = 72.68°, 22.59°, and 16.19° which are assigned to (204), (002), and (111), respectively, indicating the amorphous properties of the OPEFB [38]. From the XRD results, the PCL as a semi-crystalline polymer is shown to have three diffraction peaks around 21.42°, 22.1°, and 23.8° which are corresponding to (110), (111), and (200) respectively. The broad halo peak is mostly due to the amorphous nature of the PCL. On the other hand, there is several characteristic peak positions appear for NiO at 2θ = 79.38°, 75.20°, 62.87°, 43.20°, and 37.20° corresponding’s to (222), (311), (220), (200), and (111) respectively. The XRD data for nickel oxide shows a strong and sharp diffraction peak and the absence of diffraction halo in the recorded XRD pattern indicates the presence of crystalline phase only and the lack of any amorphous or crystalline-amorphous phase formation, respectively [23]. The XRD data for OPEFB/NiO/PCL composites show the intensity of the peaks from the amorphous and crystalline components, where many small peaks have been observed throughout the angles for the XRD patterns of the composites at different compositions. The intensity for PCL sharp peaks is reduced and the sharp peak of the fiber has disappeared indicating that the NiO peaks strongly appear when the increment of filler. From the results, the crystallinity of the materials has been increased after NiO added. This leads to changing dramatically and controlling the dielectric properties of the composites by manipulating the filler ratio [23].

### 4.2. Fourier Transforms Infrared (FT-IR) Spectroscopy

The technique of FT-IR was employed in this study to demonstrate the bonding, interaction, and monitor the shifts in absorption peaks in specific IR regions for the determination of the known interactions of the functional group of the PCL with NiO and OPEFF. Figure 5 presents the FT-IR spectra of pure material and the OPEFF/PCL, NiO/PCL and OPEFF/NiO/PCL composites with many regions of specific stretching at room temperature. FT-IR spectra of OPEFF/PCL which reveal the distinct broad peak of hydroxyl group absorption at around 3448–3440 cm^−1^. The O–H stretching vibration of cellulose, lignin component of OPEFF, hemicelluloses and absorbed water are responsible for this. The stretching vibration C–H in polycaprolactone could explain the presence of the peak around 2941.82 cm^−1^. The peak of absorption at around 1721 cm^−1^ and the peak around 1600 cm^−1^ resulting from water absorption by cellulose is an indication of the presence of carbonyl groups in lignin and hemicelluloses of OPEFF. The bending of –CH_2_ and –CH_3_ in all FT-IR spectra is indicated by the peak around 1454–1442 cm^−1^ [39]. C–O stretching of aliphatic secondary and primary alcohols in fiber is indicated by the absorption band in the region of 1167.59–1154 cm^−1^ in OPEFF spectrum [40]. An interaction between the PCL and fiber is being revealed by the absence of substantial changes in the peaks’ position. The significant peaks of OPEFF fiber was not spotted in the fiber-reinforced PCL composites because they may have been enveloped by other strong peaks of PCL, given that the magnitude of fiber was less compared to that of the PCL matrix [41]. The presence of these peaks at 1154–1167.59 cm^−1^ indicated that the fiber had been introduced into the polymer matrix. Nickel oxide (NiO) was used to reinforce the OPEFF/PCL composites. Figure 6 shows the spectra of 25 wt. % fiber-reinforced at 50 wt. % PCL with 25 wt. % NiO. There was a shift of broadband (~3444.30–3440.74 cm^−1^) to the left which corresponds to the OH stretching. This phenomenon occurred because of the existence of hetero-associated hydrogen bonds that formed between the carboxylic acid groups of PCLs and the NiO bonded group. Hydrogen bonding was the only interfacial force that facilitated the interconnection between the PCL matrix and the NiO reinforcement network.

### 4.3. Thermal Analysis (DTG and TGA) Properties of Materials

The differential thermogravimetry (DTG) and thermogravimetric analysis (TGA) thermograms results for non-composites pure materials and OPEFB/PCL, NiO/PCL, NiO/OPEFB/PCL composites are presented in Figure 6a,b, and Figure 7 respectively. Figure 6a,b indicates that the DTG was performed for temperature changes compared with the weight loss rate of different filler contents on the NiO/OPEFB/PCL composites. Weight loss for the NiO/OPEFB/PCL composites was decreased compared with those OPEFB/PCL, and NiO/PCL composites. This reduction indicated that the presence of NiO eventually assisted the formation of bonds between molecules in the NiO/OPEFB/PCL composites. Therefore, the gradual increase in the thermal stability caused an increase in the dielectric properties of the composites due to the enhancement of the bond among the constituents of the composites. 

The thermal stability of pure PCL, pure OPEFF, as well as PCL based composites, were investigated and compared in terms of weight loss as a function of temperature using thermogravimetric analysis (TGA) as shown in Figure 7. The characteristic thermal parameters selected were onset temperature, which is the initial weight loss temperature, and maximum degradation temperature (Tmax), which is the highest thermal degradation rate temperature. The thermal degradation for the pure compounds showed that the PCL has higher thermal stability than OPEFF because the onset degradation temperature peak was at 392.17 °C and it was completely decomposed at 471.83 °C, while OPEFF showed peak degradation at 250.83 °C and was fully degraded at 365.66 °C as shown in Figure 7. The onset degradation temperatures are 303.83 °C, 401.64 °C, and 303.83 for the PCL/OPEFB, NiO/PCL, and NiO/OPEFB/PCL respectively, are higher than the onset degradation temperatures of the pure OPEFF, While the (Tmax) thermal degradation temperatures are 427.50 °C, 471.67 °C, and 480.50 °C for the PCL/OPEFB, NiO/PCL, and NiO/OPEFB/PCL respectively. The improved thermal stability is attributed to the presence of OPEFF, and NiO dispersed in the polymer matrix. This reassembling creates a protective physical barrier on the surface of the material which hinders the permeability of volatile degradation products out from the blend and eventually helps delay the degradation of the blend. 

As can be seen from the Figure 7, there are two main degradation regions for the PCL-based composites. Insignificant loss in weight was observed at the first one of the degradations, this is because of which was the evaporation of moisture that occurred during the drying process of the specimens. The second degradation of weight loss attributes to both phases of the matrix’s complete degradation of the NiO/OPEFB/PCL composites, which then converts into carbon and hydrocarbon residues. Finally, the weight loss which changed with temperature change became insignificant because of the total matrix degradation into carbonaceous products as the temperature reached 600 °C. The content of nickel oxide inside the composites was not degraded as shown in Figure 7. Therefore, natural fibers added Leads to the decreased thermal stability of polymer composites. This was caused by a portion of the PCL polymer matrix that had been replaced by natural fibers, which have low thermal stability [42,43]. On the other hand, it is possible to control the thermal stability, whether an increase or decrease, depending on the type of fiber added natural or synthetic [44]. 

### 4.4. Structural Characterization of the Prepared Substrates

Scanning electron microscopy was used to study the surface morphology of the 12.2 wt. % OPEFF/87.8 wt. % PCL, 12.2 wt. % NiO/87.8 wt. % PCL, and 25 wt. % OPEFF/25 wt. % NiO/50 wt. % PCL composites. SEM Morphological images with 750× magnification shows of the cross-section for all composites are shown in Figure 8a–c. It appears that their dispersion of fiber and nickel oxide could be seen inside the matrix. Figure 8a illustrates a scanning electron micrograph of the 12.2 wt. % OPEFF/87.8 wt. % PCL surface showing the fiber surfaces are shown clean which indicated poor adhesion between the fibers and the PCL matrix. The micrographs reveal that the fibers are oriented in the form of single fibers, hence implying that the fibers have been dispersed during the blending process [45]. Good dispersion of single fibers and good fiber orientation shall contribute to producing good mechanical properties and thus result in good tensile strength for this biocomposites material [46]. Figure 8b with 750× magnification shows NiO particles were well distributed in the PCL polymer matrix indicating the presence interaction between the NiO particles and the PCL matrix. Figure 8c illustrates a scanning electron micrograph of the surface for 25 wt. % OPEFF/25 wt. % NiO/50 wt. % PCL composites. At first sight, the dispersion of NiO particles and OPEFF in the PCL matrix by the efficient mixing of NiO particles and OPEFF in the matrix via Thermo Haake blending machine. It is observed that the NiO particles in composites were randomly arranged with OPEFF in the PCL matrix, the composite showed some gaps and good interfacial.

### 4.5. Dielectric Properties of Fabricated Substrates

The variation in the dielectric constant $\varepsilon_{r}^{\prime }\left(f\right),$ loss factor $\varepsilon_{r}^{\prime \prime }\left(f\right),$ and loss tangent (tan δ) with frequency for all materials under study are illustrated in Figure 9a–c. The dielectric and loss factor for all the samples mostly decreased with the frequency. The decrease in the dielectric constant and loss factor with the frequency is due to the decrease in orientation polarization at high frequencies [45]. At higher frequencies the movement of charge cannot keep up with the alternating field, and the polarization mechanism ceases to contribute to the polarization of the dielectric. Hence as the frequency increase, the material net polarization drops since each polarization mechanism depends on contribution and this give rise to drops in dielectric constant [46]. Furthermore, the rapid rotation of the dielectric polar molecules was insufficient to attain equilibrium with the field [47].

The OPEFF has a lower dielectric constant $\varepsilon_{r}^{\prime }\left(f\right)$ and loss factor $\varepsilon_{r}^{\prime \prime }\left(f\right)$ than PCL as shown in Figure 9a,b. Thus, the increment of the percentage of OPEFF would weaken the polarizability strength of OPEFF/PCL composites, which in turn would reduce both the dielectric constant $\varepsilon_{r}^{\prime }$ and loss factor $\varepsilon_{r}^{\prime \prime }$. However, the measured $\varepsilon_{r}^{\prime }\left(f\right)$ and $\varepsilon_{r}^{\prime \prime }\left(f\right)$ for nickel oxide and nickel oxide composites are rather high as compared to OPEFF, PCL, and OPEFF/PCL substrate composite. Accordingly, the measurement results showed that polymer-fiber, nickel oxide composites exhibit controllable permittivity dielectric constant $\varepsilon_{r}^{\prime }\left(f\right)$ between 1.89 and 4.2 F/m, with loss factor $\varepsilon_{r}^{\prime \prime }\left(f\right)$ between 0.08 and 0.62 F/m, and loss tangent (tan δ) between 0.05 and 0.18 depending on the mixing ratio of filler powders to the polymer matrix as shown in Figure 9c.

### 4.6. Comparison Between S-Parameters Measurement and Simulation

Characterizations of the constitutive electromagnetic properties of the fabricated substrates were tested using the CSTMWS software package. This software package was used in order to validate the retrieved complex permittivity from the OEC technique. Some simulations were performed inside a rectangular waveguide with the same specifications and dimensions of the one used in the measurements. The TE10 mode is excited by assigning wave ports. The input complex permittivity of the sample is assumed to be the same value retrieved from the open-ended coaxial probe. The evaluated results, in terms of S-parameters, from the simulations are extracted from 8 to 12 GHz. An intensive convergence criterion with minimum errors between two consecutive iterations is assumed to be no more than 1%.

The reflection coefficient (S11) and transmission coefficient (S21) values’ variation with frequency for various substrates that have been prepared from composites of polymer placed at rectangular waveguide as well as simulation results are presented in Figure 10a–c. The X-axis is a representation of the frequency from 8–12 GHz, while the Y-axis represents a magnitude S11 and S21 at dB. The two main curves in the diagrams represents the reflection coefficient (S11) and transmission coefficient (S21) and the sum of curve values, which is near unity. An acceptable agreement can be seen from the results. Also, the fabricated substrates were used to mount T-junctions to test their S-parameters. As seen in Figure 11a–c, the simulated results show an excellent agreement with the measured results

### 4.7. Antenna Performance Measurement

#### 4.7.1. Return Loss (R.L.)

For the time being it is a prevalent practice to evaluate the system performances by computer simulation before the real-time enforcement. A simulator “Computer Simulation Technology Microwave Studio (CSTMWS) based on the finite-element method has been used to calculate bandwidth, gains, and return loss. This simulator also assistance to reduce the manufacturing cost because only the antenna with the best performance would be fabricated [48]. Definition of the Reflection loss or Return loss which is the reflection of signal power (Pr) to the incident of signal (Pi) through the device by the transmission line. It is expressed as a ratio in dB relative to the transmitted signal power. The return loss of signal is calculated by [49].
Return Loss (R.L) = 10log (Pr/Pi)(3)

The change in substrate content of composites composition (wt. %) in the patch antennas drops the resonant frequency (*fo*) to lower frequencies in both experimental and simulated values of return loss for all patch antennas, as shown in Figure 12. This was resulted from a consequential NiO and OPEFF increase which changes the dielectric constant of the substrate. The observed results are an indication of possibility of developing an antenna that can operates in 1.7–1.85 GHz frequency range, controlling only the resonator composition. Therefore, *fo* and *ε_r_,* depend on the kind of material and percentage of filler concentration in the composite [50]. The maximum RL value for NiO/OPEFB/PCL sample reaches −16.3 dB at 1.724 GHz, while the RL value for the NiO/PCL sample was obtained reaches −14.2 dB at 1.80 GHz. In addition, the minimum RL value for OPEFB/PCL sample reaches -11.93 dB at 1.79 GHz. It is clear from the investigation that measured and simulated results are likely the same that means these antennas had not many losses while transmitting the signals and the error for all samples was acceptable and as if virtually non-existent, but some fabrication losses and environmental conditions slightly altered the results.

#### 4.7.2. Radiation Pattern

The antenna performance is completely characterized in terms of radiation patterns using CSTMWS numerical environments as seen in Figure 13a–c. Figure 13 shows the radiation pattern of the all antennas understudy, OPEFF/PCL, NiO/PCL, and OPEFF/NiO/PCL at the resonant frequency of 1.95, 1.8, and 1.724 GHz, respectively. The radiation pattern of the three patch antennas is generally its most basic requirement because it determines the distribution of radiated energy in space, where It is observed a remarkable focusing in the radiated fields from the dipoles to the end of the path [51]. The purpose of this work is to design an antenna that is small-sized, consumes less power with low cost, and can be used for low band communication applications. Therefore, from the obtained results, it is expected that NiO/OPEFB/PCL structures synthesized in this work could be used as a reference to design novel microstrip patch antennas utilize at microwave applications. The antennas’ performances are illustrated carefully in Table 3. the antenna performance is completely characterized in terms of reflection (S_11_) and radiation patterns using CSTMWS numerical environments. As seen in Figure 12 for return loss and Figure 13 for the 3-D radiation patterns at 1.79 GHz, 1.80 GHz, and 1.724 GHz are presented. is observed a remarkable focusing in the radiated fields from the dipoles to the end-fire. Such enhancement is attributed to the structures of the proposed substrates which include the NiO particles played a crucial in improving and determining the dielectric and thermal properties of the substrate due to its hierarchical and porous architectures.

## 5. Conclusions and Future Remarks

The manuscript successfully studied the recycle waste organic fiber by preparing semi-flexible substrates for the purpose of employing them for the design microstrip patch antennas for low band communication applications at low cost. A systematic approach, based on experimental, analytical and numerical techniques, was used to investigate the main structure of the fabricated semi-flexible substrates from oil palm empty fruit fiber (OPEFF) and nickel oxide (NiO) nanoparticles hosted in polycaprolactone (PCL) polymer as a matrix. The morphology, crystalized structure, thermal stability behavior, and dielectric properties of substrates are studied. Three microstrip patch antenna circuitries were fabricated based on the prepared substrates and then started the testing of dielectric and reflection coefficient parameters. It was found that the measured results were in agreement with the simulated results, with a negligible error. The excellent agreement achieved between the simulated and measured results showed the validity of the proposed approach. The experimental results showed that that polymer-fiber, nickel oxide composites exhibit controllable permittivity dielectric constant $\varepsilon_{r}^{\prime }\left(f\right)$ between 1.89 and 4.2 F/m, with loss factor $\varepsilon_{r}^{\prime \prime }\left(f\right)$ between 0.08 and 0.62 F/m, and tangent tan δ between 0.05 and 0.18. Return losses measurement of the three patch antennas OPEFF/PCL, NiO/PCL, and OPEFF/NiO/PCL are (−11.93, −14.2, and −16.3) dB respectively. The evaluated results from the simulations, in terms of S-parameters, extracted from (8–12) GHz for T-junction and (1.5–2) GHz for the microstrip patch antenna design, shows that an intensive convergence criterion with minimum errors between two consecutive iterations is assumed to be no more than 1%. Important recommendations of considered future applications of natural fibers were the modification, and control over the quality of natural fibers can revolutionize their commercialization. On the other hand, the addition of other fillers in either nano-size can overcome some of the natural fiber reinforced composites’ limitations, such as the addition of multi-wall carbon nanotubes (MWCNT), reduced graphene oxide (RCO), and oxides metals these dopants might improve the structure, surface morphology, and dielectric characteristics of the prepared composites. It is worth mentioning that natural fiber-based hybrid composites bring a competitive market for various industrial applications.

## Figures and Tables

**Figure 1 polymers-12-02400-f001:**
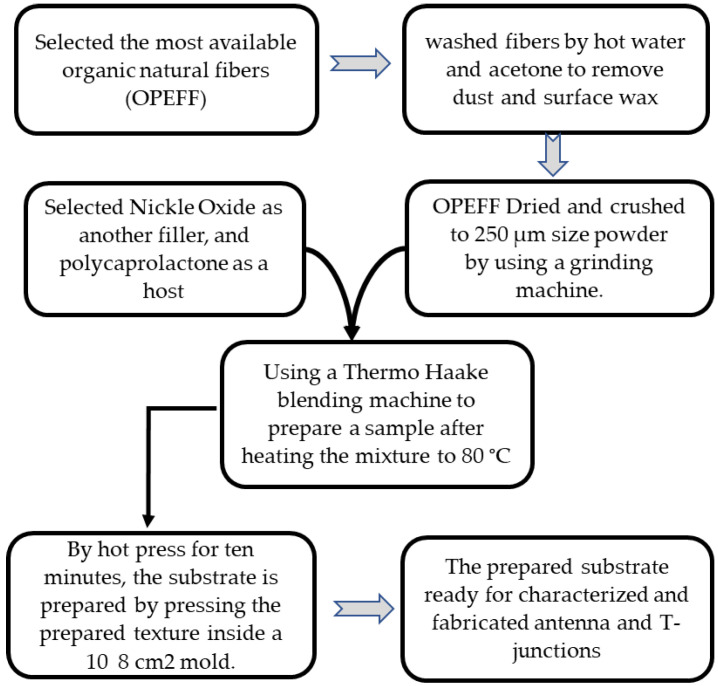
The substrates preparation procedures.

**Figure 2 polymers-12-02400-f002:**
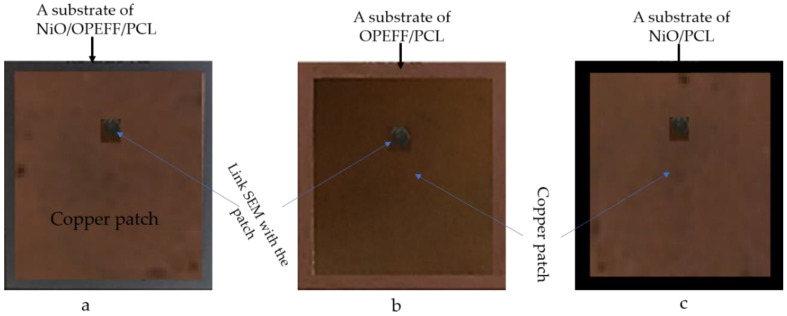
Manufactured microstrip patch antennas (**a**) NiO/oil palm empty fruit fiber (OPEFF)/polycaprolactone (PCL), (**b**) OPEFF/PCL, and (**c**) NiO/PCL based on the fabricated substrates.

**Figure 3 polymers-12-02400-f003:**
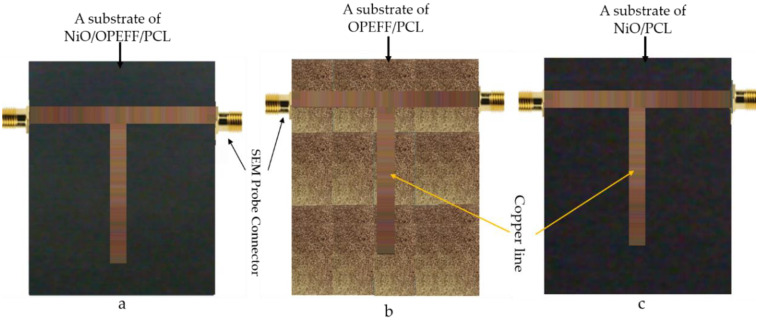
Manufactured structure of T-junctions (**a**) NiO/OPEFF/PCL, (**b**) OPEFF/PCL, and (**c**) NiO/PCL based on the fabricated substrates.

**Figure 4 polymers-12-02400-f004:**
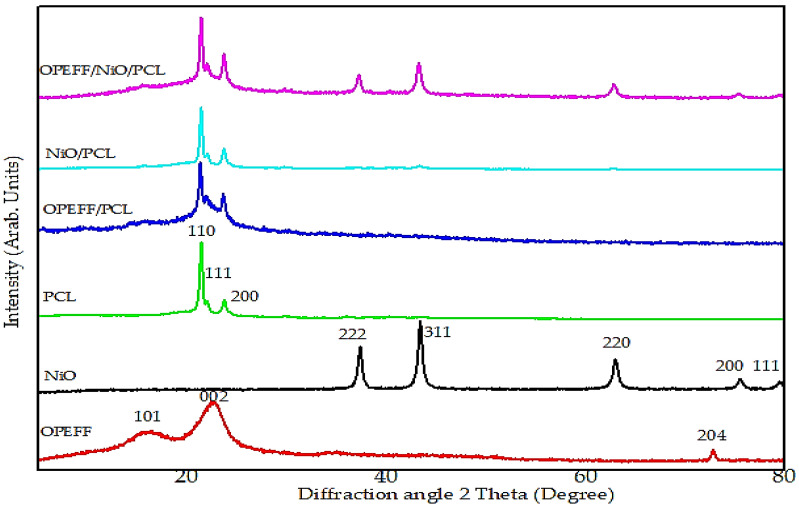
XRD patterns for pure material under test and OPEFF/NiO/PCL composites at different compositions.

**Figure 5 polymers-12-02400-f005:**
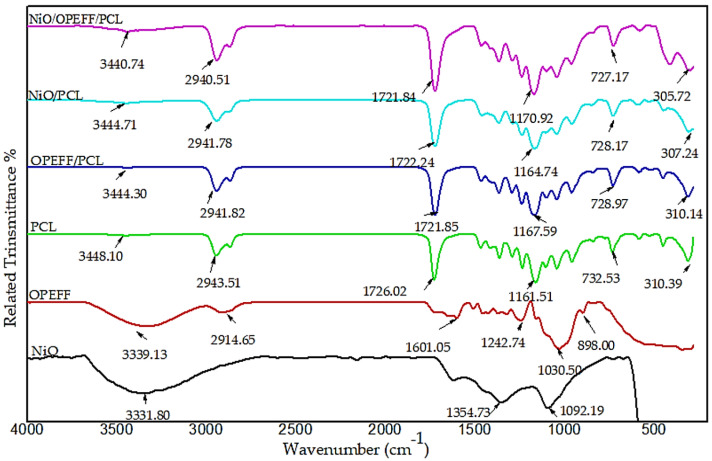
Comparison of the FT-IR spectra for all materials and composites under study.

**Figure 6 polymers-12-02400-f006:**
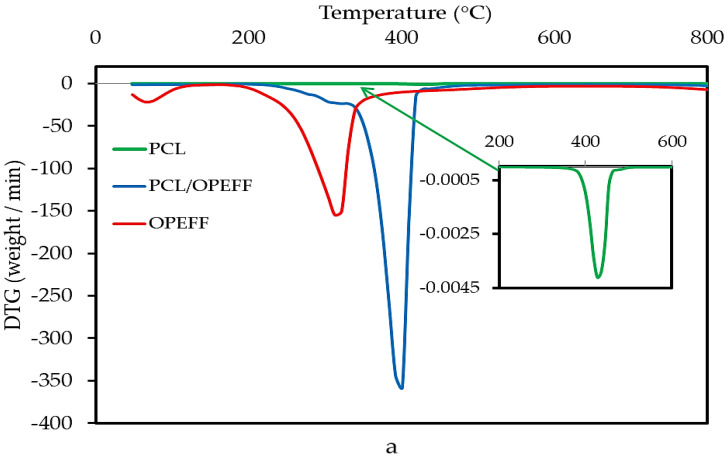
Derivative thermogravimetric (DTG) thermograms of (**a**) neat PCL, neat OPEFF, and PCL/OPEFF composite, and (**b**) Neat NiO, PCL/NiO, and PCL/NiO/OPEFF composite.

**Figure 7 polymers-12-02400-f007:**
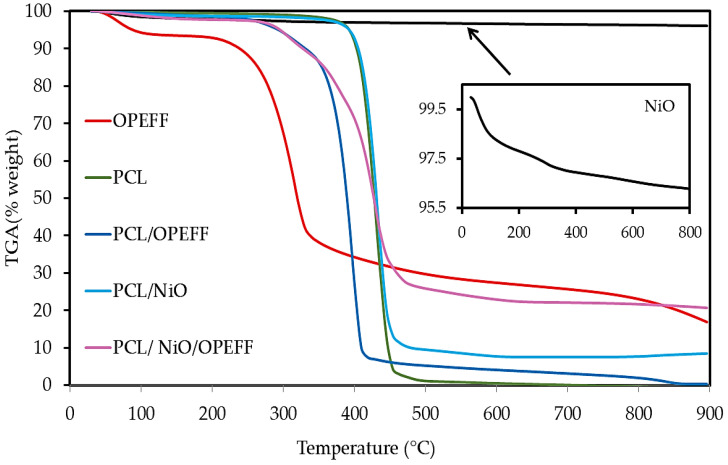
Thermogravimetric analysis (TGA) thermograms of neat PCL, neat OPEFF, neat NiO, PCL/OPEFF, PCL/NiO, PLA/NiO/OPEFF composites.

**Figure 8 polymers-12-02400-f008:**
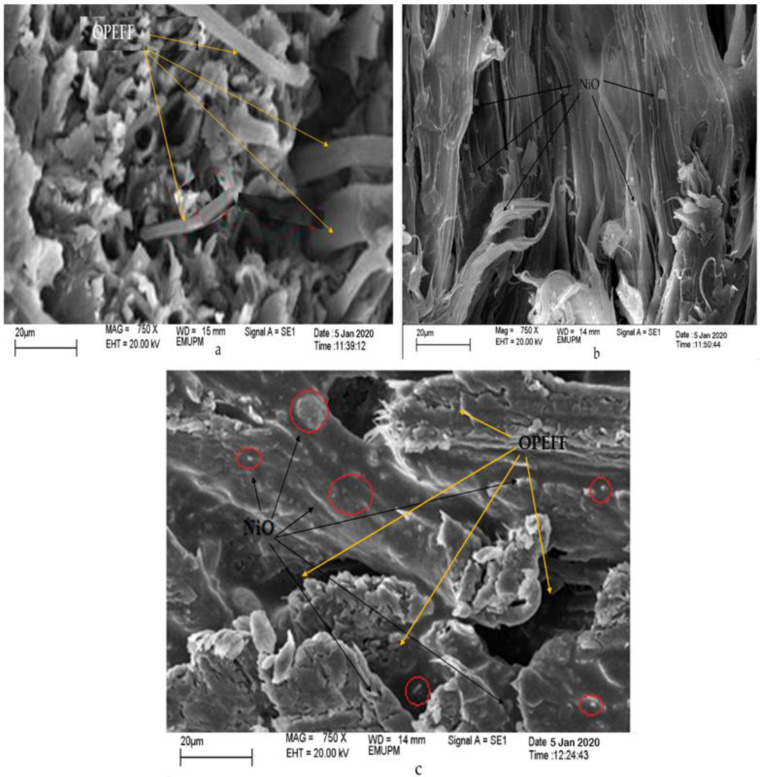
SEM micrographs of cross-sectional surfaces of (**a**) OPEFB/PCL, (**b**) NiO/PCL, and (**c**) NiO/OPEFB/PCL composites.

**Figure 9 polymers-12-02400-f009:**
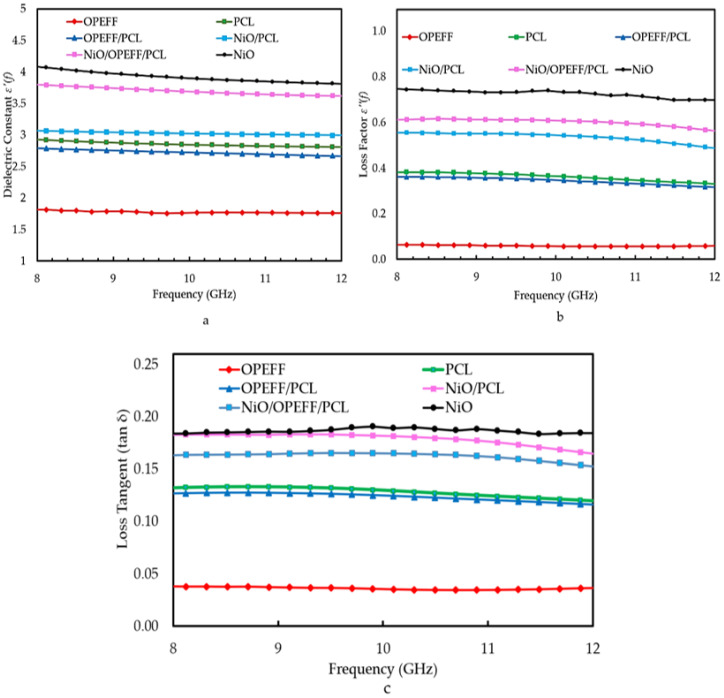
Dielectric properties (**a**) Dielectric constant (**b**) loss factor (**c**) Loss tangent of the substrate material under test OPEFF, NiO, PCL, 12.2%OPEFF/87.8%PCL, 12.2%NiO/87.8%PCL, and 25%OPEFF/25%NiO/50%PCL.

**Figure 10 polymers-12-02400-f010:**
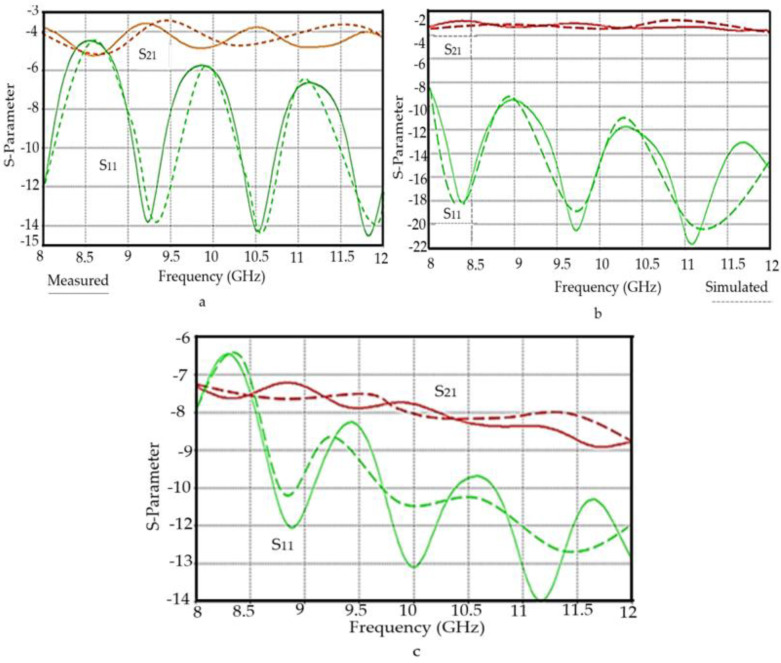
The S-parameters of the substrates (**a**) OPEFF/PCL, (**b**) NiO/PCL, and (**c**) OPEFF/NiO/PCL obtained by rectangular waveguide connected to VNA.

**Figure 11 polymers-12-02400-f011:**
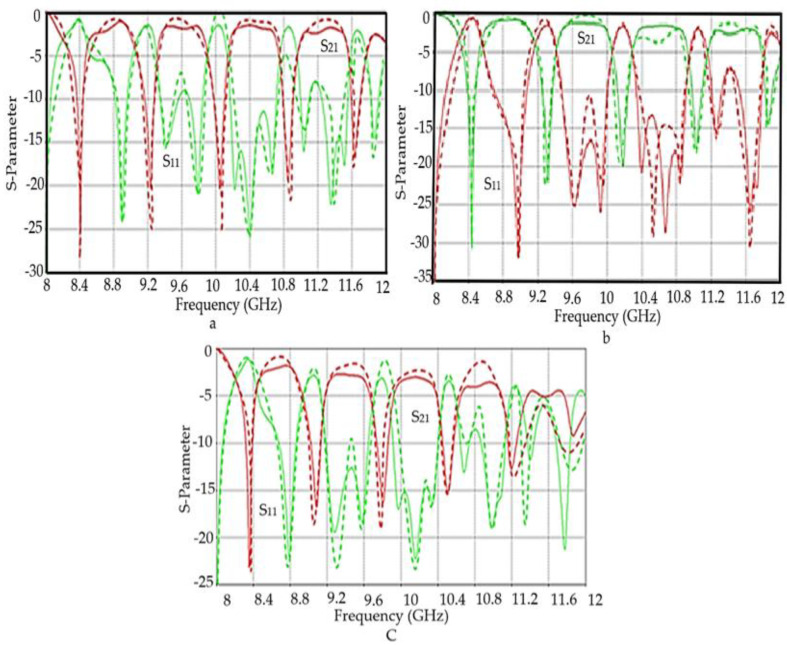
The S-parameters of (**a**) OPEFF/PCL, (**b**) NiO/PCL, and (**c**) OPEFF/NiO/PCL obtained by microstrip T-Junctions connected to VNA.

**Figure 12 polymers-12-02400-f012:**
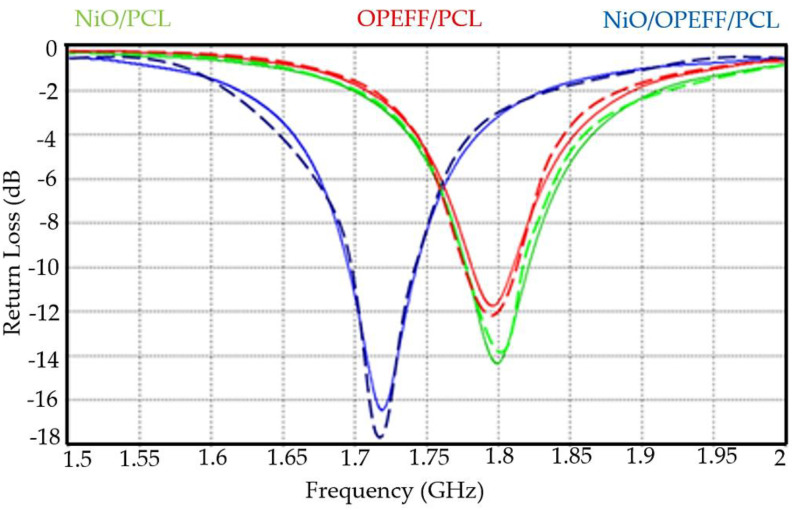
The return losses measurement and simulations results of the three microstrip antennas based on the different fabricated substrates.

**Figure 13 polymers-12-02400-f013:**
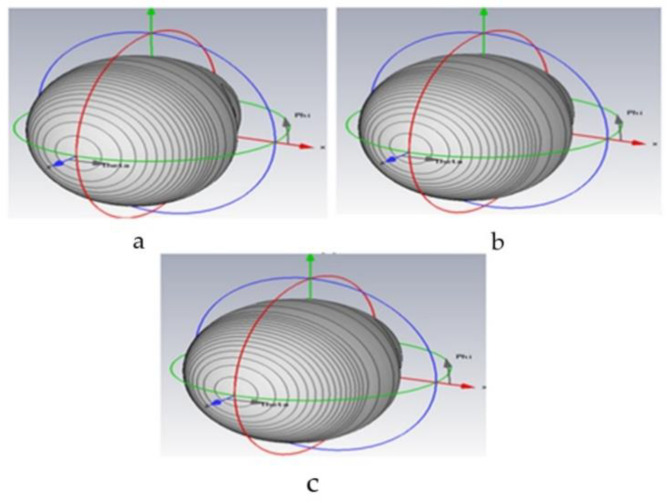
Antennas performance (**a**) 3-D Radiation pattern at 1.79 GHz for OPEFF/PCL, (**b**) 3-D Radiation pattern at 1.80 GHz for NiO/PCL, and (**c**) 3-D Radiation pattern at 1.724 GHz for NiO/OPEFF/PCL.

**Table 1 polymers-12-02400-t001:** Composition of three substrates prepared.

NiO wt. %	OPEFF wt. %	PCL wt. %
-	12.2	87.8
12.2	-	87.8
25	25	50

**Table 2 polymers-12-02400-t002:** Geometrical dimensions for all antennas.

Dimensions	(mm)
Thickness of substrate	4.35
Thickness of patch	0.05
Patch feed line	4.4
Patch size (L × W)	62 × 62

**Table 3 polymers-12-02400-t003:** Antennas performance.

Substrates Content	Return Loos (RL) dB	Resonance Frequency GHz
12.2 wt. % OPEFF/87.8 wt. % PCL	−11.93	1.79
12.2 wt. % NiO/87.8 wt. % PCL	−14.2	1.80
25 wt. % OPEFF/25 wt. % NiO/50 wt. % PCL	−16.3	1.724

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
