# Peer review of "Preparation and Characterization of Semi-Flexible Substrates from Natural Fiber/Nickel Oxide/Polycaprolactone Composite for Microstrip Patch Antenna Circuitries for Microwave Applications"

_polymers, 2020, doi:10.3390/polym12102400_

Round 1
Reviewer 1 Report
This paper presents the results of the research work focused on the preparation and characterization of the PCL based composites. The PCL/nickel oxide materials are filled with oil palm fibers. While I have no objection to the comprehensive description of the tests on dielectric properties, TGA and FTIR, the missing element of material analysis are the measurements of mechanical and structural properties. I suggest supplementing the research with the results of static bending and impact toughness measurements as well as SEM structural observations.
Author Response
Thank you for reviewing our submission and your very valuable comments. We carefully responded to all the points and have modified the manuscript accordingly. Please see it in the attachment.

Reviewer 2 Report
Manuscript: Preparation and Characterization of Semi-Flexible Substrate from Natural Fiber/ Nickel Oxide/Polycaprolactone Composite for Microstrip Patch Antenna Circuitries for Microwave Applications
The manuscript presents very good research related to use of Natural Fibers Microwave Applications and required minor correction.
- Abstract should contain some quantitative information also.
- English must be improved.
- Novelty of the work be established.
- All the results reported be compared in a tabular form to establish the superiority of the work
- Quality of figure 6 need to improve.
- Authors must need to incorporate some recent references in the introduction part of the manuscript related to the other application of natural fibers; For example.
- Cellulose 25 (3), 1961-1973 (b)
- ACS Appl. Mater. Interfaces2019, 11, 23, 21166–21176
- Natural Fibers, Plastics and Composites pp 249-274
- https://doi.org/10.1016/j.compositesb.2019.106956
- Biomacromolecules 18 (8), 2333-2342
- ACS omega 4 (26), 22008-22020
- ACS Sustainable Chemistry & Engineering 7 (6), 6140-6151
- Quality of figure 3 need to improve.
- Label important weight losses in the TGA.
- Please include future prospective of the presented work in the conclusion part of the manuscript.
Author Response
List of Responses to the Reviewer(s) Comments on the Manuscript (polymers-904843)
Preparation and Characterization of Semi-Flexible Substrate from Natural Fiber/ Nickel Oxide/Polycaprolactone Composite for Microstrip Patch Antenna Circuitries for Microwave Applications
Thank you for reviewing our submission and your very valuable comments. We carefully responded to all the points and have modified the manuscript accordingly.
(B)-The comments and suggestions for the second reviewer.
- The manuscript presents very good research related to the use of Natural Fibers Microwave Applications and required minor correction.
(X)- English language and style, Moderate English changes required.
- In general, the manuscript has been checked for linguistic, content, references, and formatting to achieve the publishing requirements of your journal.
(X)-The research introduction, design, methods, results presented and the conclusions must be improved
- The research introduction, design, results presented, and conclusions have been modified appropriately to achieve the publishing requirements of your journal.
(X)- Abstract should contain some quantitative information also.
- The abstract has been modified
(X)- English language must be improved.
- In general, the manuscript has been checked for linguistic and formatting to achieve the publishing requirements of your journal.
(X)- Novelty of the work be established.
- The novelty of this work, manufacturing a novel microstrip patch antenna circuitries based on a Semi-Flexible Substrate from Natural Fiber/ Nickel Oxide/Polycaprolactone composite for microwave applications.
- All the results reported be compared in a tabular form to establish the superiority of the work
- Table 3 has been added with more explanation to the (4.7.2. Radiation pattern) section.
(X)- Quality of figure 6 need to improve.
- The quality of figure 6 has been improved, Figure 6 was separated into two parts. Figure 6(a, and b) represents (DTG) analysis, while Figure 7 represents (TGA)analysis.
(X)- Authors must need to incorporate some recent references in the introduction part of the manuscript related to the other application of natural fibers;
- All recommended references have been added to the introduction part of the manuscript
For example.
- Cellulose 25 (3), 1961-1973 (b) Done
- ACS Appl. Mater. Interfaces2019, 11, 23, 21166–21176. Done
- Natural Fibers, Plastics and Composites pp 249-274. Done
- https://doi.org/10.1016/j.compositesb.2019.106956. Done
- Biomacromolecules 18 (8), 2333-2342. Done.
- ACS omega 4 (26), 22008-22020. Done
- ACS Sustainable Chemistry & Engineering 7 (6), 6140-6151. Done
(X)- Quality of figure 3 need to improve.
- The quality of figures 2 and 3 has been improved. In addition, figure 6 has been divided into two figures (6-7), as well as figure 8 has been added to support the SEM section explanation.
(x)- Label important weight losses in the TGA.
- Section (TGA) has been reformulated, the important weight losses thermal parameters have been added to the TGA section characterization. The onset temperature, which is the initial weight loss temperature, and maximum degradation temperature (Tmax), which is the highest thermal degradation rate temperature. Another information has been added as illustrated in the section ''thermal Analysis (DTG and TGA) Properties of materials''
(x)- Please include future prospective of the presented work in the conclusion part of the manuscript.
- The future prospects of the presented work have been added to the conclusion part of the manuscript.
’’Important recommendations of considered future applications of natural fibers, were the modification, and control over the quality of natural fibers, can revolutionize their commercialization. On the other hand, the addition of other fillers in either micro or nano-size can overcome some of the natural fiber reinforced composites' limitations. such as the addition of multi-wall carbon nanotubes (MWCNT), reduced graphene oxide (RCO), and oxides metals these dopants might improve the structure, surface morphology and dielectric characteristics of the prepared composites. It is worth mentioning that natural fiber-based hybrid composites bring a competitive market for various industrial applications.’’
If you need further clarification or modification, you are more than welcome to contact me at “ahmad_al67@yahoo.com” or ahmadfahad@upm.edu.my.
- All the comments requested by the reviewers were acted upon.
Thank You.
Ahmad Fahad Ahmad (Ph.D)
Physics Department, Faculty of Science,
Universiti Putra Malaysia
ahmad_al67@yahoo.com, ahmadfahad@upm.edu.my.

Reviewer 3 Report
In Introduction dental and orthopaedic applications, ref.4, was mistakenly cited as publication related to electronic market.
Ref.8 in the line 62 was not devoted to alleviation of environmental pollution as it was stated in the Introduction text.
In line 87 the publication by Mithelesh et al was cited but not found in references.
The content of publication cited in line 101, ref.20, differs from the content in the text in lines 97-101 of Introduction.
The grammatical errors in references 18 and 20 must be corrected.
The preparation and characterization of polymer composites for microwave applications has been described.
Author Response
List of Responses to the Reviewer(s) Comments on the Manuscript (polymers-904843)
Preparation and Characterization of Semi-Flexible Substrate from Natural Fiber/ Nickel Oxide/Polycaprolactone Composite for Microstrip Patch Antenna Circuitries for Microwave Applications
Thank you for reviewing our submission and your very valuable comments. We carefully responded to all the points and have modified the manuscript accordingly.
(C)-The comments and suggestions for the third reviewer.
- The preparation and characterization of polymer composites for microwave applications have been described.
(X)- English language and style, Moderate English changes required.
- In general, the manuscript has been checked for linguistic, content, references, and formatting to achieve the publishing requirements of your journal.
(X)-The research introduction, design, methods, results presented and the conclusions must be improved
- The research introduction, design, results presented, and conclusions have been modified appropriately to achieve the publishing requirements of your journal.
(X)- In Introduction dental and orthopedic applications, ref.4, was mistakenly cited as publication related to the electronic market.
- Ref .4. has been replaced with ‘’Ahmad, A.F.; Aziz, S.A.; Obaiys, S.J.; Zaid, M.H.M.; Matori, K.A.; Samikannu, K.; Aliyu, U.S.A. Biodegradable Poly (lactic acid)/Poly (ethylene glycol) Reinforced Multi-Walled Carbon Nanotube Nanocomposite Fabrication, Characterization, Properties, and Applications. Polymers, 2020, 12, p.427. doi; org/10.3390/polym12020427’’.
(X)- Ref.8 in line 62 was not devoted to the alleviation of environmental pollution as it was stated in the introduction text.
- Ref .8. has been replaced with’’ Mochane, M.J.; Mokhena, T.C.; Mokhothu, T.H.; Mtibe, A.; Sadiku, E.R.; Ray, S.S.; Ibrahim, I.D.; Daramola, O. O. Recent progress on natural fiber hybrid composites for advanced applications: A review. EXPRESS Polym. Lett. 2019, 13, 159–198, doi: 10.3144/ expresspolymlett.2019.15.
(X)- In line 87 the publication by Mithilesh et al was cited but not found in references.
- The reference has been changed to ‘’Islam, M.T.; Ullah, M.H.; Singh, M.J.; Faruque, M.R.I. A new metasurface superstrate structure for antenna performance enhancement. Materials, 2013, 6, 3226-3240. doi: org/ 10.3390 /ma 608 3226’’
- Note: Some references sites have been changed due to adding new references
(X)- The content of the publication cited inline 101, ref.20, differs from the content in the text in lines 97-101 of the Introduction.
- The reference (20) of the publication has become ref 27, has been replaced with Kakumani, A.D.; Ruthramurthy, B.; Wong, H.Y.; Ong, B.H.; Tan, K.B.; Yow, H.K. Microstructure and Dielectric Properties of Nickel‐Doped Ba0.7Sr0. 3TiO3 Ceramics Fabricated by Sol-gel Method. Int. J. Appl. Ceram. Technol. 2016. 13, 177-184. doi: org/10.1111/ijac.1242
(X)- The grammatical errors in references 18 and 20 must be corrected.
- The grammatical errors in both references 18 and 20 have been corrected.
If you need further clarification or modification, you are more than welcome to contact me at “ahmad_al67@yahoo.com” or ahmadfahad@upm.edu.my.
- All the comments requested by the reviewers were acted upon.
Thank You.
Ahmad Fahad Ahmad (Ph.D)
Physics Department, Faculty of Science,
Universiti Putra Malaysia
ahmad_al67@yahoo.com, ahmadfahad@upm.edu.my.

This manuscript is a resubmission of an earlier submission. The following is a list of the peer review reports and author responses from that submission.
Round 1
Reviewer 1 Report
In its current shape, the article does not meet the requirements of a scientific publication, in my opinion the presented research shows a low level of novelty. This is particularly evident when one considers the previous work of the authors, where such material systems have already been studied.
Reviewer 2 Report
The paper entitled “Fabrication and Characterization of a New Substrate Made of Natural Fiber/ Nickel Oxide/Polycaprolactone Composite for Microwave Antenna Circuitries for Wi-Fi Applications” is an interesting manuscript I suggest the publication in Polymers.
In this research activity, the authors proposed the use of recycled organic materials waste to realize semi-flexible substrates. The developed systems can be applied in low band communication applications. The main structure of the substrates was fabricated by Oil Palm Empty Fruit Fiber (OPEFF) and Nickel Oxide (NiO) nanoparticles hosted in Polycaprolactone (PCL) has been done by using thermal Haake blending machine, which ensured mixture homogeneity. The characteristic of different substrates were analyzed using different instruments: X-ray diffraction (XRD), Fourier transforms infrared spectrometry (FTIR), differential thermogravimetry and thermogravimetric analysis (TGA, DTG). The XRD results showed that the substrate profile patterns changed as the NiO loading amount was increased. FTIR spectra illustrated a slight change in the frequencies and positions of the peaks after adding NiO. The dielectric properties were analysed using an open-ended coaxial probe connected with Agilent N5230A PNA-L Network Analyzer at the frequency band of interest (8-12) GHz. The experimental results showed that the complex permittivity of the composites OPEFF/PCL is (2.76-0.316), NiO/PCL (3.07-j0.512) and OPEFF/NiO/PCL (3.80-j0.576), respectively. Return losses measurement of the three patch antennas OPEFF/PCL, NiO/PCL, and OPEFF/NiO/PCL are (-11.9, -14.2, and -16.3) dB respectively. Finally, the commercial software package, Computer Simulation Technology Microwave Studio (CSTMWS), was used to investigate the antenna performance by simulate S-parameters based on the measured dielectric parameters. The simulated and measured results were in excellent agreement but the environmental conditions slightly altered the results.
Specific comments:
The authors are invited to improve the English style
Line 106-107: The authors are invited to modify the following sentence “…the to detect improvements results. and Finally comparing the simulated results with the measured values.”
Line 116: The authors are invited to modify :
The three major components of the substrates utilized at this work are the fillers, reinforcement, and polymer.
With
The three major components of the substrates utilized in this work are the fillers, reinforcement, and polymer.
Figure 6: The authors are invited to comment better the presence of OPEFF and NiO in PCL.